# Efficacy of 1-Year Cavacurmin^®^ Therapy in Reducing Prostate Growth in Men Suffering from Lower Urinary Tract Symptoms

**DOI:** 10.3390/jcm12041689

**Published:** 2023-02-20

**Authors:** Giulio Milanese, Edoardo Agostini, Maria Vittoria De Angelis, Eugenio Pretore, Andrea Benedetto Galosi, Daniele Castellani

**Affiliations:** 1Urology Unit, ASUR Area Vasta 5, Mazzoni Hospital, 63100 Ascoli Piceno, Italy; 2Faculty of Medicine, School of Urology, Polytechinic University of Le Marche, 60121 Ancona, Italy; 3Urology Unit, Azienda Ospedaliero-Universitaria delle Marche, 60121 Ancona, Italy

**Keywords:** prostatic hyperplasia, lower urinary tract symptoms, curcumin, phytotherapy

## Abstract

We aim to assess the effect of Cavacurmin^®^ on prostate volume (PV), lower urinary tract symptoms (LUTS) and micturition parameters in men after 1 year of therapy. From September 2020 to October 2021, data from 20 men with LUTS/benign prostatic hyperplasia and PV ≥40 mL who were on therapy with α1-adrenoceptor antagonists plus Cavacurmin^®^ were retrospectively compared with 20 men on only α1-adrenoceptor antagonists. Patients were evaluated at baseline and after 1 year using the International Prostate Symptom Score (IPSS), prostate-specific antigen (PSA), maximum urinary flow (Qmax) and PV. A Mann–Whitney U-test and Chi-square were used to assess the difference between the two groups. A comparison of paired data was performed with the Wilcoxon signed-rank test. Statistical significance was set at *p*-value < 0.05. There was no statistically significant difference in baseline characteristics between the two groups. At the 1-year follow-up, PV [55.0 (15.0) vs. 62.5 (18.0) mL, *p* = 0.04)], PSA [2.5 (1.5) ng/mL vs. 3.05 (2.7) vs. *p* = 0.009] and IPSS [13.5 (3.75) vs. 18 (9.25) *p* = 0.009] were significantly lower in the Cavacurmin^®^ group. Qmax was significantly higher in the Cavacurmin^®^ group [15.85 (2.9) vs. 14.5 (4.2), *p* = 0.022]. PV was reduced to 2 (5.75) mL in the Cavacurmin^®^ group from baseline, while it increased to 12 (6.75) mL in the α1-adrenoceptor antagonists group (*p* < 0.001). PSA decreased in the Cavacurmin^®^ group [−0.45 (0.55) ng/mL], whereas it increased in the α1-adrenoceptor antagonists group [0.5 (0.30) ng/mL, *p* < 0.001]. In conclusion, one-year Cavacurmin^®^ therapy was able to block prostate growth with a concomitant decrease in PSA value from baseline. The association of Cavacurmin^®^ with α1-adrenoceptor antagonists had a more beneficial effect compared to patients on α1-adrenoceptor antagonists alone but this needs further larger studies to be confirmed, particularly in the long-term.

## 1. Introduction

Benign prostatic enlargement (BPE) due to benign prostatic hyperplasia (BPH) is the most frequent cause of lower urinary tract symptoms (LUTS) in men aged 50 years and older [1], and its prevalence increases with aging [2]. Moderate-to-severe LUTS due to BPE/BPH is a common complaint and has an important impact on the quality of life in elderly men [3], who seek medical care for LUTS treatment almost five times more likely than younger men [4].

According to current international guidelines, pharmacological treatment of BPE-LUTS is indicated in men with moderate-to-severe LUTS, particularly in those bothered by their symptoms [5]. α1-adrenoceptor antagonists and 5α-reductase inhibitors (5-ARIs) are the most frequently prescribed drugs, alone or in combination. α1-adrenoceptor antagonists are usually first-line therapy due to their rapid onset of action in reducing both storage and voiding LUTS [6]. 5-ARIs are usually indicated in men who have moderate-to-severe LUTS and an increased risk of disease progression (e.g., prostate volume >40 mL) [7]. However, both medications are frequently associated with side effects such as dizziness, orthostatic hypotension and ejaculatory dysfunction (for α1-adrenoceptor antagonists), erectile dysfunction, reduced libido, gynecomastia, and decreased semen volume (for 5-ARIs) [8].

Phytotherapy encompasses a large number of plant extracts such as pollen, roots, bark, seeds or fruits and has become a popular BPH/LUTS treatment in several countries such as Germany, France and Austria, where market shares of plant extracts reach up to 30–50% due to their safety profile and low side effects [9]. ß-sitosterols, lectins, phytosterols and fatty acids are the most important constituents identified in plant extracts used in BPH/LUTS therapy. Based on in vitro studies, numerous biological activities such as 5a-reductase inhibition, androgen blockade aromatase activity, inhibition of prostaglandin synthesis, a1-blockade and anti-inflammatory activity have been postulated as mechanisms of action in BPH [10]. Indeed, BPH/LUTS etiology is thought to be related to stimulation of prostate cell growth by dihydrotestosterone converted from testosterone by 5α-reductase, interactions between stroma and epithelium with the production of growth factors, and increased inflammation [11].

Curcumin is a fat-soluble hydrophobic polyphenol phytoextract with orange-yellow pigment derived from the root of the Indian plant turmeric (Curcuma longa L.), an aromatic rhizome of the ginger family (Zingiberaceae) [12]. Curcumin is commonly used as a food additive or natural coloring agent in Asia [12]. Curcumin has been demonstrated in in vitro and clinical studies to have anti-inflammatory, anti-oxidant, anti-viral, anti-lipidemic, antiproliferative and proapoptotic effects [13].

In a rat study, the Curcumin group demonstrated a decreased expression of growth factors (VEGF, TGF-ß1, and IGF1) in the prostatic tissue and a significantly lower prostate weight and volume compared to the testosterone-induced BPH group [14]. In another animal study, the effect of Curcumin was confirmed in vivo on reducing prostate size and the levels of proinflammatory cytokines (IL-6 and TNF-a) and exhibited in vitro attenuated prostatic cell proliferation [15]. Therefore, Curcumin may inhibit the effects of androgens and inflammation of the prostate. Nevertheless, Curcumin extract molecules are hydrophobic and therefore agglomerate in the human body. Consequently, Curcumin molecules have poor absorption, poor bioavailability, rapid metabolism and rapid elimination [12,13]. Cavacurmin^®^ (Wacker Biotech GmbH, München, Germany) is curcumin powder that disperses easily in aqueous systems and then is absorbed directly into blood vessels 10 to 40 times more compared to pure curcumin powder or commercial curcumin supplement products, showing very high oral bioavailability [16].

The present study aimed to assess the effect of Cavacurmin^®^ on prostate volume (PV) growth in men after 1 year of continuative therapy. Secondary outcomes were improvement in LUTS and micturition parameters.

## 2. Materials and Methods

A retrospective analysis was conducted on all men with LUTS who were referred to our outpatient clinic from September 2020 to October 2021. Inclusion criteria were men aged 50 years and older complaining of LUTS due to BPH who were on therapy with α1-adrenoceptor antagonists for at least 1 year and PV ≥ 40 mL. Exclusion criteria were active urinary tract infections, suspicion of prostate cancer, therapy with 5-ARIs or plant extracts within three months of presentation, PV < 40 mL, previous acute urinary retention episodes, urethral stricture, neurogenic bladder, and prior lower urinary tract surgery or pelvic radiotherapy. The following data were gathered: self-administered International Prostate Symptom Score (IPSS), prostate-specific antigen (PSA), maximum urinary flow (Qmax) at uroflowmetry, and abdominal ultrasound evaluation of post-void residual urine volume (PVR) and PV measured on transrectal ultrasound using the ellipsoid formula (length × height × width × [π/6]).

Patients were divided into two groups. Patients in group 1 were suggested to carry on taking α1-adrenoceptor antagonists, whilst patients in group 2 were additionally prescribed Qurmin 1200 mg (Naturneed Srl, Macerata, Italy), a nutraceutical component containing Cavacurmin^®^, once daily. The Qurmin prescription was per the attending physician’s choice. Patients were evaluated at 1-year follow-up, gathering the same data. The first 20 patients in each group who met inclusion criteria were included in the analysis. The primary study outcome was the evaluation of the difference in PV at one-year follow-up between the two groups. Secondary outcomes were the differences in IPSS, PSA, and micturition parameters.

Formal ethics committee approval was deemed unnecessary for this type of study because retrospective data collection was performed for clinical purposes, and all the procedures were performed as part of routine care. The study was conducted following the 1964 Helsinki declaration and its later amendments. All patients signed an informed consent to gather their anonymized data.

### Statistical Analysis

Continuous variables are presented as median and interquartile ranges. The Mann–Whitney U-test was used to assess the difference between the two groups for continuous variables and the Chi-square test for categorical ones. Comparison of paired data was performed with the Wilcoxon signed-rank test. Comparison of paired data was performed with the Wilcoxon signed-rank test. Statistical significance was set at a 2-tailed *p*-value < 0.05. Statistical tests were conducted using SPSS software package version 25.0 (IBM Corp., Armonk, NY, USA).

## 3. Results

Table 1 shows patients’ baseline characteristics. There were no statistically significant differences in baseline characteristics between the two study groups. At 1-year follow-up (Table 2), PV was significantly lower in group 2 [55.0 (15.0) vs. 62.5 (18.0) mL, *p* = 0.04)], and PSA was significantly higher in group 1 [3.05 (2.7) vs. 2.5 (1.5) ng/mL, *p* = 0.009]. IPSS was significantly lower in group 2 [13.5 (3.75) vs. 18 (9.25), *p* = 0.009], whilst Qmax was significantly higher in group 2 [15.85 (2.9) vs. 14.5 (4.2), *p* = 0.022] even if this was not clinically meaningful.

Table 3 and Figure 1 show differences from the baseline value at 1-year follow-up. Patients in group 1 demonstrated a median increase in PV of 12 (6.75) mL; conversely, PV was reduced to 2 (5.75) mL in group 2 (*p* < 0.001). Similarly, PSA decreased from baseline in group 2 (−0.45 (0.55) ng/mL), whereas it slightly increased in group 1 (0.5 (0.30) ng/mL) and the difference was statistically significant (*p* < 0.001). Regarding LUTS, there was a median IPSS reduction of −2 (3) points, whereas patients in group 1 had a median increase of 3.5 (1.75) (*p* < 0.001). No adverse events were recorded in both groups. Three patients in group 1 were offered transurethral resection of the prostate for LUTS worsening, whereas all patients in group 2 prolonged medical therapy.

## 4. Discussion

In addition to conventional pharmacological therapy, the number of men interested in or currently using phytotherapy for BPH/LUTS as alternative and/or complementary treatments is increasing steadily worldwide [17]. The rational use of plant extracts in LUTS/BPH relies, at least in vitro, on (i) anti-inflammatory, anti-androgenic and estrogenic effects; (ii) inhibition of lipoxygenase, aromatase, growth factor-stimulated proliferation of prostatic cells, α1-adrenoceptor, 5α-reductase, and vanilloid receptors; (iii) reduction of sexual hormone binding globulin; (iv) neutralization of free radicals [9,17,18,19].

In the present study, we retrospectively evaluated the efficacy of a 1-year lasting combination therapy of α1-adrenoceptor antagonists plus Cavacurmin^®^ for 20 men suffering from LUTS/BPH compared to 20 men on α1-adrenoceptor antagonists alone. We found that PV of patients on combination therapy (i.e., Cavacurmin^®^ plus α1-adrenoceptor antagonists) did not increase significantly at 1 year. Conversely, there was a median decrease of 2 mL against an increase of 12 mL in patients just on α1-adrenoceptor antagonists. This result could be explained by the fact that Curcumin may act as a 5-ARI in humans too. Kim et al. demonstrated in a rat BPH model that 4 weeks of Curcumin administration had a similar effect on PV reduction (i.e., decrease in prostate weight) and histopathologic morphology compared to finasteride treatment (i.e., decreases in hyperplasia and epithelial layer thickness), whereas non-treated rats developed prostate growth with typical BPH histology [14]. These findings suggest that Curcumin inhibits the development of BPH as finasteride does. The authors also found that the expression of VEGF, IGF1 and TGF-ß1 were decreased in the Curcumin group [14]. Previous research has demonstrated that these growth factors are associated with angiogenic growth [20], stromal extracellular matrix accumulation [21] and prostatic enlargement [22] in animal BPH models. Another animal study by Liu et al. found that treatment with Curcumin reduced PV by reduction of the epithelial–mesenchymal transition on histology [15].

The anti-androgen effect of Curcumin has been also demonstrated in men. Ide et al. randomized 85 men with a negative for cancer prostate biopsy to a daily supplement containing isoflavones and curcumin or placebo for 6 months [23]. PSA was evaluated before and at the end of treatment. The authors found that PSA levels significantly decreased in the Curcumin group, with a greater decrease in men with baseline PSA values >10 ng/mL, demonstrating that Curcumin suppresses PSA production. This finding was confirmed in our study too, where patients taking Curcumin had significantly lower PSA values after 1 year of therapy (2.5 vs. 3.5 ng/mL, *p* = 0.009). Compared to the baseline value, patients on Curcumin exhibited a decrease in PSA, whereas patients on only α1-adrenoceptor antagonists exhibited an increase, and the difference was statistically significant (−0.45 vs. +0.50 ng/mL, *p* < 0.001).

Inflammatory cells infiltrating the prostatic epithelium, stroma and ductal lumen are commonly found in the prostates of aging men [24] and inflammation is considered another key factor in BPH development [24]. Regardless of the causes of prostatic inflammation, the latter is characterized by a large infiltration of T-cells and the expression of a variety of pro-inflammatory cytokines [24,25]. Liu et al. demonstrated that administration of Curcumin in mice caused a lower expression of pro-inflammatory cytokines such as IL-6 and TNF-a as compared to the control group, indicating that Curcumin alleviates inflammation in an induced BPH mouse model [15]. The role of inflammation in BPH was also confirmed in three randomized clinical trials where non-steroidal anti-inflammatory drugs were administered in patients with LUTS/BPH in comparison with placebo for 4–24 weeks [26]. Pooled data from 183 men indicate that men on non-steroidal anti-inflammatory drugs showed a statistically significant reduction in IPSS (−2.89 points 95% CI −3.84 to −1.95, *p* < 0.001) and increase in Qmax (0.89 mL/s 95% CI 0.21–1.58, *p* = 0.01) [26]. We also found a median decrease of IPSS of 2 points in group 2, whereas patients in group 1 demonstrated a worsening of their LUTS, with a median increase of 3.5 points in their IPSS (*p* < 0.001). LUTS improvement after 1 year of Curcumin treatment may be caused by long-lasting anti-inflammatory action. Indeed, the anti-inflammatory effect of Curcumin could explain the results of a recent study where Qiao et al. randomized 122 BPH patients to receive tamsulosin 0.2 mg, finasteride 5 mg and curcumin 2250 mg once a day versus tamsulosin 0.2 mg, finasteride 5 mg and placebo [27]. At the 6-month follow-up, both groups demonstrated LUTS improvement from baseline, but IPSS in the Curcumin group decreased significantly compared with those on placebo [27].

In addition, several LUTS/BPH clinical trials confirmed a positive correlation between prostatic inflammation with an increased risk of LUTS worsening, risk of urinary retention and need for surgery [28]. This result may explain why no patient in the Curcumin group required surgery after 1 year of follow-up in our study, whilst three patients in group 1 required transurethral resection of the prostate for LUTS progression.

This study is not devoid of limitations. Firstly, its retrospective nature has inherent bias. However, baseline parameters did not differ significantly between the two study groups and this may limit the lack of randomization. In addition, the lack of placebo in group 1 could partially explain the better IPSS score at 1-year follow-up in group 2. Nevertheless, PV did not increase in group 2 at follow-up and this might explain, at least in part, the difference in IPSS between the groups. Finally, our study results rely on a small cohort, and further prospective studies with longer follow-up are needed to confirm our exploratory results.

## 5. Conclusions

Our study shows for the first time that one-year Cavacurmin^®^ therapy was able to block prostate growth in humans with a concomitant decrease in PSA value from baseline. In addition, the association of Cavacurmin^®^ with α1-adrenoceptor antagonists successfully improved LUTS and micturition, with a significantly more beneficial effect compared to patients on α1-adrenoceptor antagonists, and no patient in the former group required surgery, whereas three patients in the latter group underwent transurethral resection of the prostate for LUTS worsening. However, the efficacy of this combination therapy needs further, larger studies to be confirmed, particularly in the long-term.

## Figures and Tables

**Figure 1 jcm-12-01689-f001:**
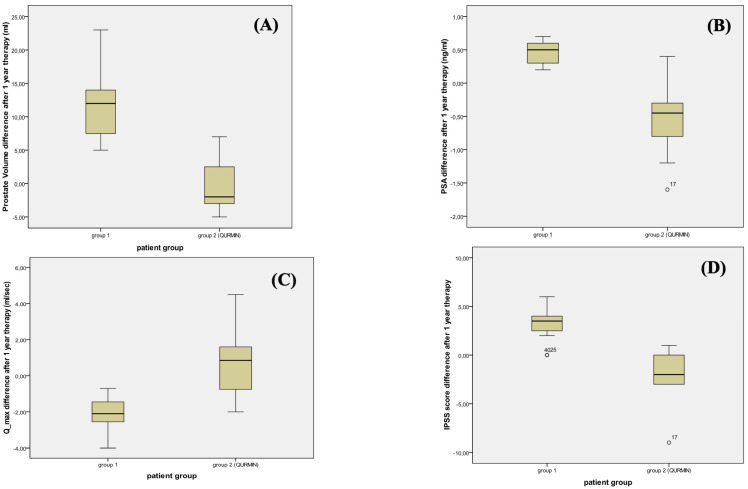
Box plot graphs of change after 1 year of therapy in both groups for (**A**) prostate volume (PV); (**B**) prostate-specific antigen (PSA); (**C**) maximum flow rate (Q-max); (**D**) international prostate symptom score (IPSS).

**Table 1 jcm-12-01689-t001:** Baseline patient characteristics. IQR: interquartile range. PSA: prostate-specific antigen. PV: prostate volume. IPSS: International Prostate Symptom Score. PVR: post-voiding residual.

	Group 1 (α1-Adrenoceptor Antagonists Only)	Group 2 (α1-Adrenoceptor Antagonists Plus Qurmin)	*p* Value
Age, years, median (IQR)	66.5 (4)	68.0 (4)	0.357
PV, mL, median (IQR)	52.0 (13)	56.00 (18)	0.379
IPSS, median (IQR)	15.5 (8)	15.0 (5)	0.958
PVR, mL, median (IQR)	62.5 (18)	55.0 (15)	0.08
Qmax, mL/s, median (IQR)	16.60 (3.3)	15.75 (2.9)	0.465
PSA, ng/mL, median (IQR)	2.60 (2.6)	3.25 (1.8)	0.828
Type of α1-adrenoceptor antagonist *n* (%)			0.81
Tamsulosin	9 (45.0)	10 (50.0)	
Alfuzosin	5 (25.0)	4 (20.0)	
Sylodosin	6 (30.0)	6 (30.0)	

**Table 2 jcm-12-01689-t002:** IQR: interquartile range. PSA: prostate-specific antigen. PV: prostate volume. IPSS: International Prostate Symptom Score. PVR: post-voiding residual.

	Group 1 (α1-Adrenoceptor Antagonists Only)	Group 2 (α1-Adrenoceptor Antagonists Plus Qurmin)	*p* Value
PV, mL, median (IQR)	62.5 (18.0)	55.0 (15.0)	0.04
IPSS, median (IQR)	18 (9.25)	13.5 (3.75)	0.009
PVR, mL, median (IQR)	60.5 (17)	56.5 (15)	0.33
Qmax, mL/s, median (IQR)	14.5 (4.2)	15.85 (2.9)	0.022
PSA, ng/mL, median (IQR)	3.05 (2.7)	2.5 (1.5)	0.009

**Table 3 jcm-12-01689-t003:** Difference from baseline value at one-year follow-up. IQR: interquartile range. PSA: prostate-specific antigen. PV: prostate volume. IPSS: International Prostate Symptom Score. PVR: post-voiding residual.

	Group 1 (α1-Adrenoceptor Antagonists Only)	Group 2 (α1-Adrenoceptor Antagonists Plus Qurmin)	*p* Value
PV, mL, median (IQR)	12 (6.75)	−2 (5.75)	<0.001
IPSS, median (IQR)	3.5 (1.75)	−2 (3)	<0.001
PVR, mL, median (IQR)	2.3 (6.9)	2.1 (7.8)	0.33
Qmax, mL/s, median (IQR)	−2.1 (1.15)	0.85 (2.52)	0.001
PSA, ng/mL, median (IQR)	0.50 (0.30)	−0.45 (0.55)	<0.001

## Data Availability

Data will be provided by the corresponding author upon reasonable request.

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
