# Peer review of "Efficacy of 1-Year Cavacurmin® Therapy in Reducing Prostate Growth in Men Suffering from Lower Urinary Tract Symptoms"

_jcm, 2023, doi:10.3390/jcm12041689_

Round 1

Reviewer 1 Report

Efficacy of 1-Year Cavacurmin® Therapy in Reducing Prostate 2 Growth in Men Suffering From Lower Urinary Tract 3 Symptoms 

Major comments:

1.rationale of the study is not appropirately defined, cite and write the background 

2. conclusions are over sighted, discuss the limitations of the study 

3. In depth justification of literature in respect to results obtained should be discussed

Author Response

Major comments:

1.rationale of the study is not appropirately defined, cite and write the background 

RESPONSE. We would like to thank you for this comment. We the greatest respect of your comment, we believe that we have already provided a well and defined contest of our study in the original submission (i.e. Introduction). Both animal and human studies were cited and discussed

  1. conclusions are over sighted, discuss the limitations of the study 

RESPONSE. We would like to thank you for this important comment. Conclusions have been implemented according to your valuable suggestion as follows “the association of Cavacurmin® with α1-adrenoceptor antagonists successfully improved LUTS and micturition with significantly more beneficial effect as compared to patients on α1-adrenoceptor antagonists, and no patient in the former group required surgery, whereas 3 patients in the latter group underwent transurethral resection of the prostate for LUTS worsening. However, the efficacy of this combination therapy needs further, larger studies to be proven, particularly in the long-term.” The limitations of our study are already present in a dedicated paragraph in the original submission as follows “This study is not devoid of limitations. Firstly, its retrospective nature with its inherent bias. However, baseline parameters did not differ significantly between the two study groups and this may limit the lack of randomization. In addition, the lack of placebo in group 1 could partially explain the better IPSS score at 1-year follow-up in group 2. Nevertheless, PV did not increase in group 2 at follow-up and this might explain, at least in part, the difference in IPSS between the groups. Finally, study results rely on a small cohort, and further prospective studies with longer follow-up are needed to confirm our exploratory results. ”

  1. In depth justification of literature in respect to results obtained should be discussed

RESPONSE. We would like to thank you for this comment. We the greatest respect of your comment, we believe that we have already discussed deeply our results compared with those of the literature. Indeed, we have cited all study which assessed Curcumin in BPH both in animal and human settings.

Reviewer 2 Report

In the Abstract, the conclusion should justify the findings from this series of cases. The present conclusion is not appropriated for a retrospective study including only 20 patients as a study group and should be rephrased.

In the Methods, the authors should provide more information:

·      More demographics details of patients included in the two study groups.

·      More details regarding the modality of intake of α1-adrenoceptor antagonists in each group.

·      Methods of data acquisition at baseline and at one-year follow-up (PV, IPSS, PSA, and micturition parameters) should be indicated.

In the Results,

·      In the sentence “IPSS was significantly lower in group 2 [13.5 (3.75) vs. 18 (9.25), p=0.009], whilst Qmax was significantly in group 2 [15.85 (2.9) vs. 14.5 (4.2), p=0.022] even if it was not clinically meaningful.”, authors should specify if Qmax value is significantly lower or higher in group 2.

In the Discussion,

·      Authors cited the study of Qiao et al. stating that “At the 6-month follow-up, both groups demonstrated LUTS improvement from baseline but IPSS in the Curcumin group decreased significantly compared with those on placebo”. How do authors explain these findings compared with their own?

In the Conclusion:

·      The conclusion should justify the findings from this series of cases. The present conclusion is not appropriated for a retrospective study including only few patients and should be rephrased.

The efficacy of one-year Cavacurmin® therapy in combination with α1-adrenoceptor antagonists needs further studies to be proven, particularly in the long-term.

Author Response

In the Abstract, the conclusion should justify the findings from this series of cases. The present conclusion is not appropriated for a retrospective study including only 20 patients as a study group and should be rephrased.

RESPONSE. We would like to thank you very much for this comment. According to your valuable suggestion, we implemented abstract conclusion as follows “The association of Cavacurmin® with α1-adrenoceptor antagonists had more beneficial effect as compared to patients on α1-adrenoceptor antagonists alone but it needs further larger studies to be proven, particularly in the long-term.”

In the Methods, the authors should provide more information:

  • More demographics details of patients included in the two study groups.

RESPONSE. We would like to thank you very much for this comment. Unfortunately, we cannot provide more details of patients because we did not gather

  • More details regarding the modality of intake of α1-adrenoceptor antagonists in each group.

RESPONSE. We would like to thank you very much for this comment. type of α1-adrenoceptor antagonists was added for each group in Table 1

  • Methods of data acquisition at baseline and at one-year follow-up (PV, IPSS, PSA, and micturition parameters) should be indicated.

RESPONSE. We would like to thank you very much for this comment. We improved this part adding that IPSS was self-administered. For the rest, information is already present in the original submission as follows “maximum urinary flow (Qmax) at uroflowmetry, abdominal ultrasound evaluation of post-void residual urine volume (PVR) and PV measured on transrectal ultrasound using the ellipsoid formula (length x height x width x [π/6]). ”

In the Results,

  • In the sentence “IPSS was significantly lower in group 2 [13.5 (3.75) vs. 18 (9.25), p=0.009], whilst Qmax was significantly in group 2 [15.85 (2.9) vs. 14.5 (4.2), p=0.022] even if it was not clinically meaningful.”, authors should specify if Qmax value is significantly lower or higher in group 2.

RESPONSE. We would like to thank you very much for this comment. We do apologize for this oversight. The sentence was rephrased as follows “IPSS was significantly lower in group 2 [13.5 (3.75) vs. 18 (9.25), p=0.009], whilst Qmax was significantly higher in group 2 [15.85 (2.9) vs. 14.5 (4.2), p=0.022] even if it was not clinically meaningful.”

In the Discussion,

  • Authors cited the study of Qiao et al. stating that “At the 6-month follow-up, both groups demonstrated LUTS improvement from baseline but IPSS in the Curcumin group decreased significantly compared with those on placebo”. How do authors explain these findings compared with their own?

RESPONSE. We would like to thank you very much for this comment. the findings of Qiao et al study was in line with our study where we found that, despite not having a placebo, the addition of Curcumin to α1-adrenoceptor antagonists improved LUTS decreasing IPSS.

In the Conclusion:

  • The conclusion should justify the findings from this series of cases. The present conclusion is not appropriated for a retrospective study including only few patients and should be rephrased. The efficacy of one-year Cavacurmin® therapy in combination with α1-adrenoceptor antagonists needs further studies to be proven, particularly in the long-term.

RESPONSE. We would like to thank you for this important comment. Conclusions have been implemented according to your valuable suggestion as follows “the association of Cavacurmin® with α1-adrenoceptor antagonists successfully improved LUTS and micturition with significantly more beneficial effect as compared to patients on α1-adrenoceptor antagonists, and no patient in the former group required surgery, whereas 3 patients in the latter group underwent transurethral resection of the prostate for LUTS worsening. However, the efficacy of this combination therapy needs further, larger studies to be proven, particularly in the long-term.”

Round 2

Reviewer 1 Report

all points are addressed